# Impacts of Outdoor Particulate Matter Exposure on the Incidence of Lung Cancer and Mortality

**DOI:** 10.3390/medicina58091159

**Published:** 2022-08-25

**Authors:** Jung-Soo Pyo, Nae Yu Kim, Dong-Wook Kang

**Affiliations:** 1Department of Pathology, Uijeongbu Eulji Medical Center, Eulji University School of Medicine, Uijeongbu-si 11759, Korea; 2Department of Internal Medicine, Uijeongbu Eulji Medical Center, Eulji University School of Medicine, Uijeongbu-si 11759, Korea; 3Department of Pathology, Chungnam National University Sejong Hospital, 20 Bodeum 7-ro, Sejong 30099, Korea; 4Department of Pathology, Chungnam National University School of Medicine, 266 Munhwa Street, Daejeon 35015, Korea

**Keywords:** lung cancer, particulate matter, meta-analysis, mortality

## Abstract

*Background and objectives*: Long-term exposure to air pollution has been associated with lung cancer. This study aimed to evaluate the relative risk (RR) and hazard ratio (HR) of lung cancers and the prognostic implication of outdoor particulate matter (PM) pollution using a meta-analysis. *Materials and Methods*: We performed the meta-analysis using 19 eligible studies and evaluated the PMs, dividing into PM smaller than 2.5 µm (PM_2.5_) and PM smaller than 10 µm (PM_10_). In addition, subgroup analyses, based on the increment of PM exposure, location, sex, smoking history, and tumor histology, were performed. *Results*: Lung cancer was significantly increased by exposure to PM_2.5_ (RR 1.172, 95% confidence interval (CI) 1.002–1.371), but not PM_10_ exposure. However, there was no significant correlation between PM_10_ exposure and the incidence of lung cancers (RR 1.062, 95% CI 0.932–1.210). The all-cause and lung-cancer-specific mortalities were significantly increased by PM_2.5_ exposure (HR 1.1.43, 95% CI 1.011–1.291 and HR 1.144, 95% CI 1.002–1.307, respectively). However, PM_10_ exposure significantly increased the all-cause mortality, but not the lung-cancer-specific mortality. The lung-cancer-specific mortality was significantly increased by PM10 per 12.1 μg/m^3^ increment and in the Europe area. *Conclusions*: PM_2.5_ significantly increased lung cancer and the all-cause and lung-cancer-specific mortalities, whereas PM_10_ did not increase lung cancer or lung-cancer-specific mortality. However, PM_10_ increased the all-cause mortality and the PM_10_ per 12.1 μg/m^3^ increment and PM_10_ in the Europe area may increase the lung-cancer-specific mortality.

## 1. Introduction

Lung cancer is one of the most frequently diagnosed cancers and the leading cause of cancer-related mortality worldwide [1]. Nonsmoking-related lung cancers, suggesting other environmental factors, such as occupational exposure and residential radon, have been implicated as causes of lung cancer [2]. In addition, long-term exposure to air pollution has been associated with lung cancer [2]. The International Agency for Research on Cancer classifies outdoor particulate matter (PM) exposure as carcinogenic to humans [3]. Because lung cancers are associated with various causes, including PM exposure, it is difficult to determine the individual effect of PM exposure on the incidence of lung cancers. In addition, regional differences may be important in studies of exposure to PM [3,4]. In various regions, long-term exposure to fine PM has been found to increase the risk of lung cancer [5,6,7,8,9,10,11]. PM is classified by particle size, specifically PM_2.5_ (fine particles with an aerodynamic diameter ≤ 2.5 μm) and PM_10_ (inhalable particles with a diameter ≤ 10 μm), and also comprises a mixture of PM_2.5_ and PM_10_, which are prominent components. Because particle size affects the degree of penetration into the lung, the harmful effects of PM can be different for different PM sizes [12]. Previous studies have reported that PM_2.5_ included a higher proportion of mutagenic species and was more carcinogenic than PM_10_ [13,14,15,16,17]. In addition, PM_10_ mainly contains minerals and biological materials [17]. A comparison of carcinogenicity between PM_2.5_ and PM_10_ may be needed to elucidate the impact of particle size on the incidence of lung cancers. 

In the present study, we aimed to evaluate the impact on the incidence of lung cancer and the prognostic implication of outdoor PM using a meta-analysis. We performed subgroup analyses based on the degree of increment, location, sex, smoking history, and tumor histology.

## 2. Materials and Methods

### 2.1. Published Study Search and Selection Criteria

The literature search was performed using the PubMed database on 15 May 2022. The search was performed using the following keywords: “particulate matter” and “lung” and “carcinoma”. The titles and abstracts of searched articles were primarily screened for exclusion. Literature including original research and systematic review articles were also screened to identify additional eligible studies. The inclusion and exclusion criteria were as follows: (1) studies on the incidence of lung cancers and the all-cause and/or lung-cancer-specific mortality from PM exposure in humans were included and (2) non-original articles, such as case reports or review articles, were excluded.

### 2.2. Data Extraction

For the meta-analysis, data were extracted from the eligible studies as follows [2,5,18,19,20,21,22,23,24,25,26,27,28,29,30,31,32,33,34]: the first author’s name, study location, study year, number of patients analyzed, and mean concentration of exposed PM. The hazard ratio (HR) and its confidence interval (CI) for the exposure to PM were investigated from eligible studies. In addition, the relative risk (RR) of incidence of lung cancers due to PM exposure was investigated.

### 2.3. Statistical Analyses

In the present meta-analysis, all data were analyzed and obtained using the Comprehensive Meta-Analysis software package Ver. 2 (Biostat, Englewood, NJ, USA). The RRs and HRs after PM exposure were determined and used in the meta-analysis. We performed subgroup analysis based on the increment of PM exposure, study location, sex, smoking history, and tumor histology. In this meta-analysis, the interpretation of the fixed and random effect models used the values of a random-effects model. The heterogeneity between eligible studies was assessed using Q and I^2^ statistics and presented using *p*-values. In addition, sensitivity analysis was conducted to assess the heterogeneity of eligible studies and the impact of each study on the combined effect. To assess the publication bias, Egger’s test was used. If significant publication bias was found, the fail-safe N and trim-fill tests were performed to confirm the degree of publication bias. A *p*-value < 0.05 was considered significant using a two-sided analysis.

## 3. Results

### 3.1. Selection and Characteristics of Studies

A total of 375 studies were identified in the PubMed database search for the meta-analysis. Through the primary screening, 338 studies were excluded. Full-text reviews were undertaken for the remaining 37 studies. Finally, 19 studies were selected according to the inclusion and exclusion criteria. Among excluded studies, 170 studies were excluded due to a lack of sufficient information, such as the incidence of lung cancers and the all-cause and/or lung-cancer-specific mortality from PM exposure. In addition, 96 studies were excluded due to non-human studies. The remaining reports were excluded for the following reasons: non-original articles (*n* = 68), focusing on other diseases (*n* = 11), articles in a language other than English (*n* = 7), and duplicate articles (*n* = 4) (Figure 1). The characteristics of the eligible studies are shown in Table 1.

### 3.2. The Incidence of Lung Cancers by PM Exposure

The RRs of lung cancers due to exposure to PM_2.5_ and PM_10_ were 1.081 (95% CI 0.939–1.245) and 0.972 (95% CI 0.914–1.034), respectively (Table 2). There were no significant differences in the RRs of lung cancers due to exposure to PM_2.5_ and PM_10_. The incidence of lung cancer was non-significantly increased by the exposure of PM_2.5_ (per 10 μg/m^3^ increment; RR 1.081, 95% CI 0.939–1.245) and PM_10_ (per 10 μg/m^3^ increment; RR 1.062, 95% CI 0.932–1.210). In the PM_2.5_ subgroup, the RRs in Asia were significantly increased by the exposure to PM (RR 1.061, 95% CI 1.044–1.078), but not in North America (RR 1.082, 95% CI 0.853–1.372). In subgroup analysis based on smoking history, RR of lung cancers was significantly higher in the former smoker subgroup but not in the never or current smoker subgroups. The RR of lung cancers was 1.650 (95% CI 1.040–2.619) in small cell carcinomas. Among the histologic subgroup of non-small cell carcinomas, squamous cell carcinoma was significantly correlated with increased RR by the exposure of PM_2.5_ (RR 1.151, 95% CI 1.107–1.198). In the exposure of PM_10_, the RRs of lung cancers were not significantly different by smoking history and tumor histology. 

### 3.3. The Mortality by PM Exposure

The mortalities due to all causes and lung cancers were marginally increased by the exposure to PM_2.5_ (HR 1.143, 95% CI 1.011–1.291 and HR 1.144, 95% CI 1.002–1.307, respectively; Table 3). In Europe, the mortalities due to all causes and lung cancers were significantly increased by the exposure to PM_2.5_ (HR 1.010, 95% CI 1.000–1.020 and HR 1.144, 95% CI 1.002–1.307, respectively). However, in Asia, the mortality by all causes, but not lung cancer, was significantly increased. There was no significant difference in the North America subgroup. The mortality due to exposure to PM_10_ was significantly increased in the all causes subgroup but not in the lung cancer subgroup (HR 1.091, 95% CI 1.023–1.162 and HR 1.168, 95% CI 0.962–1.419, respectively). In the PM_10_ subgroup, the mortality due to lung cancers was significantly increased per 12.1 μg/m^3^ increment and in Europe (HR 1.270, 95% CI 1.250–1.290 and HR 1.930, 95% CI 1.294–2.879, respectively). However, in the PM_10_ subgroup of North America, the mortality by all causes, but not lung cancer, was significantly increased.

## 4. Discussion

Outdoor PM has been implicated as a carcinogen [15]. The impact of outdoor PM exposure can differ between location, exposure period, and outdoor activity. However, the impact of outdoor PM exposure according to other various factors is unclear. To the best of our knowledge, the present study is the first study using a meta-analysis to elucidate the factors that impact PM exposure on the incidence of lung cancers and all-cause and lung-cancer-specific mortality.

In existing studies, PM in outdoor air is generally classified into PM_2.5_ and PM_10_. Larger particles can be filtered out by the clearance system of the upper respiratory tract. Thus, smaller particles can penetrate more deeply. Because the PM size affects the degree of penetration into the lung, the harmful effects of different PM sizes can differ [12]. In addition, PM_2.5_ includes a higher proportion of mutagenic species [13,14,15,16,17]. A comparison between particle size groups will be needed to elucidate the impact of PM exposure. In addition, the concentration and exposure to PM can differ by location; thus, subgroup analysis based on locations is needed. 

Previous studies have reported correlations between PM_2.5_ exposure and the incidence and mortality of lung cancers [2,5,18,19,20,21,22,23,24,25,26,27,28,29,30,31,32,33,34]. In a previous meta-analysis, lung cancer risk was found to significantly increase, by 9%, according to 18 studies [15]. The previous meta-analysis collected data from worldwide studies from the 1970s to the 2000s. However, the means of incidence and mortality are unclear in this previous study [15]. We evaluated the impact of PM exposure on the incidence of lung cancers. In the present study, the incidences of lung cancers due to PM exposure were evaluated. The RRs of lung cancers due to exposure to PM_2.5_ and PM_10_ were 1.081 (95% CI 0.939–1.245) and 0.972 (95% CI 0.914–1.034), respectively. Although PM exposure was associated with a slight increase in the incidence of lung cancers, there was no statistical significance between the incidence of lung cancers and PM exposure. 

Hamra et al. reported the relative risks of lung cancers due to outdoor PM exposure using a meta-analysis and assessed lung cancer risk for the combined incidence and mortality of lung cancers [15]. They reported that the relative risks of lung cancer were 1.09 (95% CI 1.04–1.14) and 1.08 (95% CI 1.00–1.17) due to PM_2.5_ and PM_10_ exposure, respectively. Similar to the previous meta-analysis, we analyzed subgroups based on the degree of PM exposure (per 10 μg/m^3^ increment). Lung cancer was slightly increased by the exposure of PM_2.5_ (per 10 μg/m^3^ increment; RR 1.081, 95% CI 0.939–1.245) and PM_10_ (per 10 μg/m^3^ increment; RR 1.062, 95% CI 0.932–1.210). However, statistical significance was found. In contrast to our study, Hamra et al. obtained and converted the raw data from the authors of the original studies [15]. This factor may be the cause of the different results between the present and previous meta-analyses. 

In the present study, subgroup analyses based on various factors, including the increment of PM exposure, location, sex, smoking history, and tumor histology, were performed. In a previous meta-analysis, the RRs of lung cancers were evaluated per 10 μg/m^3^ increment [15]. We assessed the change in mortality according to the increment of PM exposure. In addition, a comparison of mortality between all causes and lung cancers after PM exposure was performed. Statistical significance was found in some increment subgroups. In the all causes subgroup, the HRs due to the exposure to PM_2.5_ were 1.194 and 1.101 per 5.3 and 10 μg/m^3^ increment, respectively. The HRs due to the exposure to PM_10_ were 1.043, 1.190, 1.070, and 1.150 per 6, 7, 10, and 29.5 μg/m^3^ increment, respectively. The increasing values were slightly lower than those from lung cancers. 

We examined detailed analyses of mortality due to PM exposure, dividing mortality into all causes and lung cancers. Following exposure to PM_2.5_ and PM_10_, mortality due to all causes was significantly increased. However, the mortality due to lung cancers increased considerably in the PM_2.5_ subgroup but not in the PM_10_ subgroup. As described above, the mutagenic species were more included in PM_2.5_ than in PM_10_ and were more carcinogenic than PM_10_ [13,14,15,16,17]. In the subgroup analysis, there was some difference in mortality between locations. In the all causes subgroup, mortalities were significantly increased in Asia and Europe. However, in North America, there was a significant increase in mortality due to PM_2.5_ exposure, but not due to PM_10_ exposure. In North America, the change in mortality due to lung cancer was similar to that due to all causes. In addition, in the PM_10_ exposure subgroup, there was a significant correlation between PM exposure and mortality by lung cancer in Europe but not in North America. 

Previous studies reported the correlation between air pollution and lung cancer in never-smokers [35,36]. In addition, no significant differences were found in the incidence of lung cancers between sex and between smoking history. Per 10 μg/m^3^ increment of PM_2.5_, the RRs of North America and Europe subgroups were 1.11 (95% CI 1.05–1.16) and 1.03 (95% CI 0.89–1.20), respectively. Per 10 μg/m^3^ increment of PM_10_, the RRs of North America and Europe subgroups were 1.02 (95% CI 0.96–1.09) and 1.27 (95% CI 0.96–1.68), respectively. Statistical significance was found only in North America with a per 10 μg/m^3^ increment of PM_2.5_. 

In Yang’s report, the incidence of overall lung cancers was significantly increased due to PM_2.5_ exposure (6.0%, 95% CI 4.3–7.7%) [5]. In addition, squamous cell carcinomas and adenocarcinomas were significantly increased due to PM_2.5_ exposure (14.8%, 95% CI 10.3–19.4%, and 6.5%, 95% CI 3.3–9.8%, respectively) [5]. Hamra et al. reported that adenocarcinomas were significantly correlated with PM exposure based on a meta-analysis [15]. However, there was no significant correlation between the incidence of squamous cell carcinomas and PM exposure [15]. Moon et al. reported that an increasing incidence of lung cancers was identified in adenocarcinomas of current smokers [2]. However, the increasing incidence of other histology types was not determined [2]. In our study, the incidences of non-small cell and small cell lung cancers were compared in contrast to previous studies. The exposure of PM_2.5_ had a significant impact on increased RR in squamous cell and small cell carcinomas. However, in the PM_10_ exposure group, there were no statistical differences in various histologic subtypes.

Some limitations in the current meta-analysis exist. First, a comparison of the mortality between our study and previous studies in tumor histology could not be performed due to insufficient information. Second, changes in the incidence of lung cancers due to the degree of PM exposure other than 10 μg/m^3^ increment could not be obtained from eligible studies. The present study only performed an analysis for the per 10 μg/m^3^ increment. Third, the impact of the mixture of PM_2.5_ and PM_10_ could not be evaluated due to a lack of information in eligible studies. Fourth, the analysis according to previous cancer history could not be included in the present meta-analysis due to insufficient information of eligible studies. Fifth, a detailed evaluation according to the pack years of cigarette consumption could not be performed due to no information on eligible studies. Six, we could not evaluate the effects of concentration range. Instead of concentration range, the impacts of the increment of PM were investigated and evaluated.

## 5. Conclusions

In conclusion, exposure to PM was significantly correlated with mortality by all causes. However, the incidence and mortality of lung cancer were significantly increased by PM_2.5_ exposure, but not PM_10_ exposure. Although various factors associated with PM exposure affect the incidence of lung cancers and mortality, careful interpretation is needed, such as the size and exposure of PM and location.

## Figures and Tables

**Figure 1 medicina-58-01159-f001:**
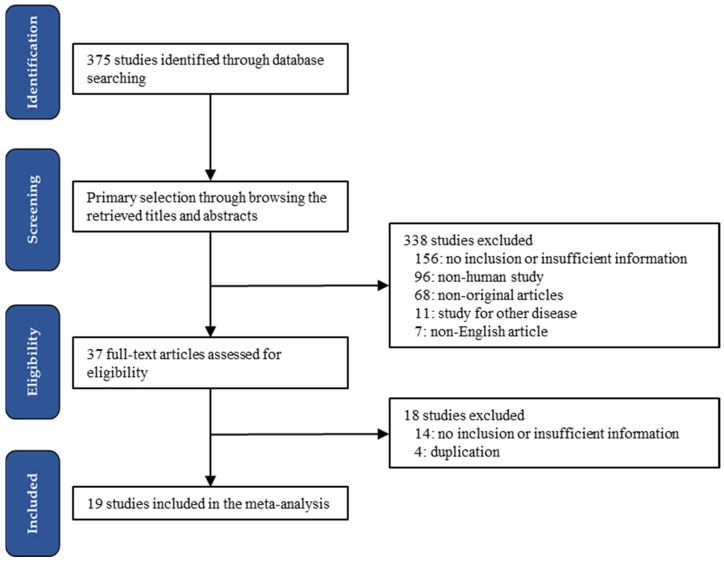
Flow chart of the searching strategy.

**Table 1 medicina-58-01159-t001:** Main characteristics of the eligible studies.

	Location	Period	Number of Patients	Subgroup	Outcome of Investigation: Concentration of PM (μg/m^3^)	
					PM_2.5_	PM_10_
Carey 2013 [18]	UK	2002	830,842		12.9 ± 1.4	19.7 ± 2.3
Cesaroni 2013 [19]	Italy	1996–2010	1,265,058		23.0 ± 4.4	NA
Eckel 2016 [20]	USA	1988–2009	352,053		13.7 ± 5.3	31.8 ± 12.1
Gharibvand 2017 [21]	USA	2000–2001	80,044	LC cases	13.11 ± 3.98	NA
				Non-LC cases	12.88 ± 3.7	NA
Gowda 2019 [22]	USA	1993–1998	65,419	LC cases	13.1 ± 2.9	NA
				Non-LC cases	13.3 ± 3.1	NA
Hart 2011 [23]	USA	1985–2000	53,814		14.1 ± 4.0	26.8 ± 6.0
Heinrich 2013 [24]	Germany	1985–1994	4752		NA	(34.8–52.5) *
Hystad 2013 [25]	Canada	1975–1994	8897		11.9 ± 3.0	NA
Jerrett 2013 [26]	USA	1998–2002	73,711		14.1 ± 12.4	NA
Katanoda 2011 [27]	Japan	1974–1983	63,520		(16.8–41.9) *	NA
Lamichhane 2017 [28]	Korea	1995–2014	1816	Adenocarcinoma	NA	55.3 ± 7.8
Lepeule 2012 [29]	USA	1979–2009	8096		15.9	NA
Lipsett 2011 [30]	USA	1996–2005	133,479		15.6 ± 4.5	29.2 ± 9.7
McDonnell 2000 [31]	USA	1973–1977	6338		31.9 ± 10.7	59.2 ± 16.8
Moon 2020 [2]	Korea	2002–2007	6,567,909		NA	55.8 ± 6.3
Pope CA 3rd 2002 [32]	USA	1979–1983	1,200,000		21.1 ± 4.6	NA
		1999–2000			14.0 ± 3.0	NA
		1982–1998			NA	28.8 ± 5.9
Puett 2014 [33]	USA	1994–2010	1,510,027		NA	NA
Tomczak 2016 [34]	Canada	1980–2005	89,835		9.1 ^†^ (1.3–17.6) *	NA
Yang 2020 [5]	China	2001–2016	12,150,000		77.3 ± 17.7	NA

PM, particulate matter; LC, lung cancer; NA, not applicable. *, range in exposure, ^†^ median.

**Table 2 medicina-58-01159-t002:** The estimated relative risk of incidence of lung cancers according to the particulate matter sizes and the subgroup analysis.

	Number of References	Heterogeneity Test (*p*-Value)	Random Effect (95% CI)	Egger’s Test (*p*-Value)
PM_2.5_	6	0.002	1.081 (0.939, 1.245)	0.848
per 10 μg/m^3^ increment	6	0.002	1.081 (0.939, 1.245)	0.848
Asia	1	1.000	1.061 (1.044, 1.078)	NA
North America	5	0.001	1.082 (0.853, 1.372)	0.831
Male	1	1.000	1.590 (1.052, 2.404)	NA
Female	2	0.423	0.973 (0.694, 1.362)	NA
Never smoker	4	0.759	1.016 (0.769, 1.340)	0.992
Former smoker	2	0.483	1.278 (1.032, 1.584)	NA
Current smoker	3	0.002	1.147 (0.790, 1.665)	0.985
Adenocarcinoma	5	0.017	1.210 (0.971, 1.508)	0.331
Squamous cell carcinoma	2	0.500	1.151 (1.107, 1.198)	NA
Large cell carcinoma	1	1.000	0.650 (0.406, 1.040)	NA
Small cell carcinoma	1	1.000	1.650 (1.040, 2.619)	NA
PM_10_	6	0.308	0.972 (0.914, 1.034)	0.489
per 10 μg/m^3^ increment	2	0.890	1.062 (0.932, 1.210)	NA
Asia	6	0.308	0.972 (0.914, 1.034)	0.489
Male	2	0.092	0.971 (0.831, 1.134)	NA
Female	4	0.517	0.990 (0.925, 1.060)	0.811
Never smoker	9	0.885	0.919 (0.871, 0.970)	0.717
Current smoker	5	0.298	0.936 (0.771, 1.137)	0.552
Adenocarcinoma	19	0.132	0.970 (0.920, 1.022)	0.337
Squamous cell carcinoma	7	0.103	0.999 (0.927, 1.076)	0.904
Large cell carcinoma	6	0.116	0.938 (0.843, 1.044)	0.461
Small cell carcinoma	6	0.879	0.860 (0.683, 1.081)	0.680

CI, confidence interval; PM, particulate matter; NA, not applicable.

**Table 3 medicina-58-01159-t003:** The estimated hazard ratios for risk of the all-cause and lung-cancer-specific mortalities according to the increasing particulate matter concentration and geographical locations.

	Number of References	Heterogeneity Test (*p*-Value)	Random Effect (95% CI)	Egger’s Test (*p*-Value)
PM_2.5_, all causes	7	<0.001	1.143 (1.011, 1.291)	0.428
per 5.3 μg/m^3^ increment	2	<0.001	1.194 (0.898, 1.587)	NA
per 10 μg/m^3^ increment	5	<0.001	1.101 (1.029, 1.178)	0.021
Asia	1	1.000	1.240 (1.121, 1.371)	NA
Europe	1	1.000	1.010 (1.000, 1.020)	NA
North America	5	<0.001	1.156 (0.989, 1.350)	0.586
PM_2.5_, lung cancer	8	<0.001	1.144 (1.002, 1.307)	0.169
per 4 μg/m^3^ increment	1	1.000	1.021 (0.950, 1.097)	NA
per 5.3 μg/m^3^ increment	2	<0.001	1.221 (0.938, 1.590)	NA
per 10 μg/m^3^ increment	7	0.055	1.166 (1.055, 1.288)	0.182
per 24.3 μg/m^3^ increment	1	1.000	2.230 (0.558, 8.910)	NA
Europe	8	<0.001	1.144 (1.002, 1.307)	0.169
PM_10_, all causes	4	0.066	1.091 (1.023, 1.162)	0.204
per 6 μg/m^3^ increment	1	1.000	1.043 (1.011, 1.077)	NA
per 7 μg/m^3^ increment	1	1.000	1.190 (1.080, 1.311)	NA
per 10 μg/m^3^ increment	1	1.000	1.070 (0.988, 1.158)	NA
per 29.5 μg/m^3^ increment	1	1.000	1.150 (0.939, 1.408)	NA
Europe	2	0.095	1.124 (1.013, 1.247)	NA
North America	2	0.351	1.045 (1.013, 1.079)	NA
PM_10_, lung cancer	5	<0.001	1.168 (0.962, 1.419)	0.491
per 6 μg/m^3^ increment	1	1.000	0.999 (0.922, 1.083)	NA
per 10 μg/m^3^ increment	2	0.001	1.307 (0.640, 2.668)	NA
per 12.1 μg/m^3^ increment	1	1.000	1.270 (1.250, 1.290)	NA
per 29.5 μg/m^3^ increment	1	1.000	1.840 (0.594, 5.704)	NA
Europe	1	1.000	1.930 (1.294, 2.879)	NA
North America	4	<0.001	1.082 (0.881, 1.328)	0.311

CI, confidence interval; PM, particulate matter; NA, not applicable.

## Data Availability

Data available in a publicly accessible repository.

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
