# Peer review of "Impacts of Outdoor Particulate Matter Exposure on the Incidence of Lung Cancer and Mortality"

_medicina, 2022, doi:10.3390/medicina58091159_

Round 1

Reviewer 1 Report

Dear Editor,

This article entitled “Impacts of Outdoor Particulate Matter Exposure on the Incidence of Lung Cancers and the Mortality” by Jung-Soo Pyo et al is an interesting paper and provides sufficient information that deserves clinicians’ attention. The authors tried to investigate the differences between PM2.5 and PM10 in relation to lung cancer regarding its incidence and mortality using meta-analysis technique. They also related those PM to the incidence of lung cancer and its prognosis by subgroup analyses according to the degree of increment, location, sex, smoking history, and tumor histology. Some points are raised to improve its quality, I hope. Please check English language if possible.

Abstract

Line 16: relative risk of ...and the hazard ratio of .... The size of ... (Please insert)

Line 18: 18 or 19 eligible studies? Please clarify.

Lines 19-20. Delete the sentence (The ……studies.).

Lines 20-21: subgroup analyses, ...., were performed. (consult English language editing)

Line 22: “increased”? (consult English language editing)

Line 31: The conclusion here is not completely encompassed all what you found. Please add.

Introduction

Lines 46-51: Please move this paragraph to line 40 or re-arrange the presentation of background of the study.

Lines 51-55: Comparison of carcinogenicity between PM2.5 and PM10 is clearly presented as shown in ref 13-17. Why needed comparison again in this study? Please explain.

Lines: 56-59: Please present the statement in order as what you presented in this manuscript. (i.e. incidence of lung cancer first, then mortality….)

Methods

Line 65: literature including original research and systematic review articles.......

2.3. statistical analysis

In meta-analysis, why do the fixed and random effect models use the values of a random-effects model? If that so, why should you present the RR value of fixed effect model in Table 2? How to interpret heterogeneity test? Will it interfere interpretation? Below Egger's test, what does the symbol “-“ mean? which was tested by fail-safe N and trim-fill tests? Please explain.

Lines 116-117: In non-small cell lung cancer, only squamous cell lung cancer was significant in relation to PM2.5. Why present non-small cell lung cancer significantly related to PM2.5? All non-small cell lung cancer about is due to squamous cell lung cancer, isn’t it?

Table 2 heading. Please change to “The estimated relative risk of incidence of lung cancer according to subgroup analysis of particulate matter.” As mentioned before, is it necessary to present fixed effect? Moreover, all are relative risk (RR) values of random effect model. Please do not present fixed effect or random effect; rather they are RRs.

Table 2: “number of subsets” is not clear. Is it “number of references”? If it is, please clarify which references that were selected.

The lines about “non-small cell………”, please delete them.

3.3.

In this section, it was not completely interpreted the data shown in table 3. Please explain.

Table 3. Please use subheadings for all causes and lung cancer by underlying (PM2.5 and PM10). Please align and make per 5.3 ug/m3 increment in one line.

4. Discussion

Line 140: “other” various factors

Lines 141-2: the first study using meta-analysis to elucidate the factors that impact PM exposure on .......... 

Lines 151-2: move to?

Lines 159-60 are the same with lines 168-9. Please do not iterate.

Line 166: meta-relative risks? What does this mean?

Line 171: "slightly increased"? it seems that the authors mean "significant".  Please reword.

Lines 173-4: Hamra et al obtained the original data form the original studies in their meta-analysis report. However, the authors of the current study did not. It seems that Hamra's study is stronger. Please explain.

Lines 194-7: Please explain why.

Lines 206-18: please why the results of the current study are inconsistent with those of previous studies?

Lines 219-20: “between our study and previous studies in tumor histology......” Please consider and change.

Lines 220-223: not clear. “…..other than 10ug/m3 increment” (inserted between exposure and could)

Lines 226-7: not clear. “of cigarette consumption” (inserted between years and could)

Lines 232-3: …is needed “such as……..”. Please explain.

Author Response

Reviewer 1.

This article entitled “Impacts of Outdoor Particulate Matter Exposure on the Incidence of Lung Cancers and the Mortality” by Jung-Soo Pyo et al is an interesting paper and provides sufficient information that deserves clinicians’ attention. The authors tried to investigate the differences between PM2.5 and PM10 in relation to lung cancer regarding its incidence and mortality using meta-analysis technique. They also related those PM to the incidence of lung cancer and its prognosis by subgroup analyses according to the degree of increment, location, sex, smoking history, and tumor histology. Some points are raised to improve its quality, I hope. Please check English language if possible.

Response:

                  We checked the English language before submission.

Abstract

Line 16: relative risk of ...and the hazard ratio of .... The size of ... (Please insert)

Response:

                  As a recommendation, we corrected the sentence.

Line 18: 18 or 19 eligible studies? Please clarify.

Response:

                  As a recommendation, we corrected the number of included studies to 19 eligible studies.

Lines 19-20. Delete the sentence (The ……studies.).

Response:

                  As a recommendation, we deleted the sentence.

Lines 20-21: subgroup analyses, ...., were performed. (consult English language editing)

Response:

                  As a recommendation, we corrected the sentence and checked the sentence.

Line 22: “increased”? (consult English language editing)

Response:

                  As a recommendation, we corrected the sentence and checked the sentence.

Line 31: The conclusion here is not completely encompassed all what you found. Please add.

Response:

                  As a recommendation, we corrected the sentence.

Introduction

Lines 46-51: Please move this paragraph to line 40 or re-arrange the presentation of background of the study.

Response:

As a recommendation, we corrected the paragraph.

Lines 51-55: Comparison of carcinogenicity between PM2.5 and PM10 is clearly presented as shown in ref 13-17. Why needed comparison again in this study? Please explain.

Response:

                  In our results, the incidences of lung cancers after the exposure of PM2.5 and PM10 were 1.081 (95% CI 0.939-1.245) and 0.972 (95% CI 0.914-1.034), respectively. There was no significant difference of the incidence of lung cancers between the exposure of PM2.5 and PM10. In the present study, the proportion of mutagenic species was not investigated.

Lines: 56-59: Please present the statement in order as what you presented in this manuscript. (i.e. incidence of lung cancer first, then mortality….)

Response:

                  As a recommendation, we corrected it.

Methods

Line 65: literature including original research and systematic review articles.......

Response:

                  As a recommendation, we corrected the sentence.

2.3. statistical analysis

In meta-analysis, why do the fixed and random effect models use the values of a random-effects model? If that so, why should you present the RR value of fixed effect model in Table 2? How to interpret heterogeneity test? Will it interfere interpretation? Below Egger's test, what does the symbol “-“ mean? which was tested by fail-safe N and trim-fill tests? Please explain.

Response:

                  Eligible studies used the different population of included patients. Therefore, the fixed effect model is not appropriate in the interpretation of results.

                  We interpret the data using the random effect model as a description. However, some readers can interpret and accept the data by evaluating the heterogeneity test (p-value), as the result of fixed or random effect models. To diminish the misunderstanding, we showed the data of fixed and random effect models. As a recommendation, we deleted the data of the fixed effect model.

                  In Egger’s test, the symbol “-” means “not applicable.” We replaced the symbol “-” to “NA.”

Lines 116-117: In non-small cell lung cancer, only squamous cell lung cancer was significant in relation to PM2.5. Why present non-small cell lung cancer significantly related to PM2.5? All non-small cell lung cancer about is due to squamous cell lung cancer, isn’t it?

Response:

                  As a recommendation, we deleted the result of non-small cell carcinoma. In addition, the data was deleted in Table 2.

Table 2 heading. Please change to “The estimated relative risk of incidence of lung cancer according to subgroup analysis of particulate matter.” As mentioned before, is it necessary to present fixed effect? Moreover, all are relative risk (RR) values of random effect model. Please do not present fixed effect or random effect; rather they are RRs.

Response:

                  As a recommendation, we corrected the sentence. In addition, we deleted the data of the fixed effect model.

Table 2: “number of subsets” is not clear. Is it “number of references”? If it is, please clarify which references that were selected.

Response:

As a recommendation, we corrected

The lines about “non-small cell………”, please delete them.

Response:

As a recommendation, we deleted it.

3.3.

In this section, it was not completely interpreted the data shown in table 3. Please explain.

Response:

As a recommendation, we added the explanations.

Table 3. Please use subheadings for all causes and lung cancer by underlying (PM2.5 and PM10). Please align and make per 5.3 ug/m3 increment in one line.

Response:

                  As a recommendation, we edited table 3.

  1. Discussion

Line 140: “other” various factors

Response:

As a recommendation, we corrected

Lines 141-2: the first study using meta-analysis to elucidate the factors that impact PM exposure on .......... 

Response:

As a recommendation, we corrected

Lines 151-2: move to?

Response:

                  We deleted the sentence.

Lines 159-60 are the same with lines 168-9. Please do not iterate.

Response:

As a recommendation, we checked and deleted it.

Line 166: meta-relative risks? What does this mean?

Response:

                  We used it as the expression of reference. To diminish misunderstanding, we corrected the term from “meta-relative risks” to “relative risks.”

Line 171: "slightly increased"? it seems that the authors mean "significant".  Please reword.

Response:

                  The meaning is “not significant.” To diminish the misunderstanding, we added the comment as “However, statistical significance was found.”

Lines 173-4: Hamra et al obtained the original data form the original studies in their meta-analysis report. However, the authors of the current study did not. It seems that Hamra's study is stronger. Please explain.

Response:

                  The included number of eligible studies is larger in the present study than in Hamar’s study. In addition, the obtained data in Hamra’s study was not the published data. Therefore, the strength cannot be evaluated between our result and Hamra’s study.

Lines 194-7: Please explain why.

Response:

                  To diminish the misunderstanding, we corrected the sentences.

Lines 206-18: please why the results of the current study are inconsistent with those of previous studies?

Response:

                 This study obtained new estimates through a meta-analysis. Various factors, including the number of population, can affect the estimates of a meta-analysis. Therefore, the results of the meta-analysis can be different from the results of previous individual studies.

Lines 219-20: “between our study and previous studies in tumor histology......” Please consider and change.

Response:

As a recommendation, we corrected the sentences.

Lines 220-223: not clear. “…..other than 10ug/m3 increment” (inserted between exposure and could)

Response:

                  As a recommendation, we corrected it.

Lines 226-7: not clear. “of cigarette consumption” (inserted between years and could)

Response:

                  As a recommendation, we corrected it.

Lines 232-3: …is needed “such as……..”. Please explain.

Response:

                  As a recommendation, we corrected it.

Reviewer 2 Report

Dear authors,

This manuscript titled ”impacts of outdoor particulate matter exposure on the incidence of lung cancers and the mortality ”, written by jung-soo pyo, nae yu lim and dong-wook kang, is very interesting and important to the health risk area of air pollutants. they used meta-analysis to find the possible effects from PM.

And there are some pointed needed to revise,

1st, please rewrite this research goal to make it more clear and accurate

2nd, about discussion part, please add some explanation about why PM2.5 is more toxic than PM10,

3rd, please show how to calculate the value of HR and CI,

4th , does the PM2.5& PM10 concentration range effect the lung cancer indence?

In my opinion, this manuscript could be considered to discuss about accept or not after these revisions.

All the best.

Author Response

Reviewer 2.

This manuscript titled ”impacts of outdoor particulate matter exposure on the incidence of lung cancers and the mortality ”, written by jung-soo pyo, nae yu lim and dong-wook kang, is very interesting and important to the health risk area of air pollutants. they used meta-analysis to find the possible effects from PM.

And there are some pointed needed to revise,

1st, please rewrite this research goal to make it more clear and accurate

Response:

                  As a recommendation, we corrected the research goal.

2nd, about discussion part, please add some explanation about why PM2.5 is more toxic than PM10,

Response:

                  As a recommendation, the explanation was added.

3rd, please show how to calculate the value of HR and CI,

Response:

                  We did not calculate the HR and CI from eligible studies. We investigated the value of HR and CI of eligible studies.

4th, does the PM2.5& PM10 concentration range effect the lung cancer incidence?

Response:

                  We could not evaluate the effects of concentration range. Instead of concentration range, the impacts of the increment of PM were investigated and evaluated. We added the description for this limitation in the revised manuscript.

In my opinion, this manuscript could be considered to discuss about accept or not after these revisions.

All the best.

Round 2

Reviewer 1 Report

I have reviewed the revised manuscript. It has been much improved; however, I have some minor comments which were written directly on the manuscript PDF. I hope this would not cause inconvenience. 

1. Title: lung cancer or cancers?
2. Abstract: please re-write according to my suggestions.
3. Heading of Tables 2 and 3 and some minor suggestions.

Author Response

I have reviewed the revised manuscript. It has been much improved; however, I have some minor comments which were written directly on the manuscript PDF. I hope this would not cause inconvenience.

  1. Title: lung cancer or cancers?

Response:

                  We confirm the term as “lung cancer.”

  1. Abstract: please re-write according to my suggestions.

Response:

                  As a recommendation, we re-write the abstract.

  1. Heading of Tables 2 and 3 and some minor suggestions.

Response:

                  As a recommendation, we corrected.